# The Moderating Role of Research and Development (R&D) Support in the Relationship between Entrepreneurship and per Capita Output—A Study on the GCC Countries

**Houcine Benlaria \*, Naeimah Fahad S. Almawishir, Sawssan Saadaoui, Sanaa Mostafa Mohammed Mohammed, Badreldin Mohamed Ahmed Abdulrahman** and **Ibrahhim Ahmed ELamin Eltahir**

College of Business, Jouf University, Sakaka 72388, Saudi Arabia; smostafa@ju.edu.sa (S.M.M.M.); iaeltahir@ju.edu.sa (I.A.E.E.)
* Correspondence: hbenlarir@ju.edu.sa

**Abstract:** The current study examines the moderating role of R&D expenditures by the government on the relationship between entrepreneurship and per capita output in GCC countries. Using secondary quantitative data, panel data analysis was conducted for six GCC countries (Saudi Arabia, Kuwait, Oman, Bahrain, Qatar, and UAE) based on their scores on the Global Entrepreneurship Index, Ease of Doing Business, and R&D expenditure as a percentage of GDP. Descriptive statistics and regression analysis were conducted using Eviews 12. The study found that a supportive business environment and entrepreneurship ecosystem can lead to higher per capita output and that laboratory force and capital are significant positive contributors to per capita output. However, both Ease of Doing Business and the Global Entrepreneurship Index have a significant negative impact on per capita output. The study did not find significant moderation of the relationship between entrepreneurship and per capita output by R&D expenditures. These findings have important implications for policy-makers and academia, emphasizing the significant labour force and capital for per capita output. Future research should explore the relationship between entrepreneurship and growth further and investigate the role of R&D. Policy recommendations include reducing regulatory burdens and providing tax incentives to create a supportive environment for entrepreneurship and increasing R&D funding to promote per capita output. Overall, this study contributes to the state of the art through examining the moderating role of R&D expenditures on the relationship between entrepreneurship and per capita output in the context of GCC countries.

**Keywords:** R&D; entrepreneurship; GCC; economic; per capita output

## 1. Introduction

According to Inekwe (2015), technology is the main foundation of the modern world, as technological advancements drive socio-economic development in any society. Continued technical innovation is necessary to fuel economic expansion. Solow's view suggests that technological advancement is external to the production function of capital and labour, and economies eventually reach a stable state with no growth in output per capita (Osiobe 2019). On the other hand, the new per capita output hypothesis highlights the importance of technological advancement in the expansion of most economies. The endogenous growth model, which is an improved growth model, is crucial for achieving long-term growth (Sharipov 2015). The interest in finding mechanisms that explain the underlying relationship between entrepreneurship and per capita output is growing, based on new endogenous growth theories that can elucidate the reasons behind income disparities between countries.

The relationship between research and development expenditure (RED), per capita output, and entrepreneurship is a topic of much discussion among business leaders, policy-makers, and researchers. While R&D expenditure is essential for promoting innovation,

the degree to which it impacts per capita output and entrepreneurship remains uncertain (Lindholm-Dahlstrand et al. 2019). Some argue that increasing R&D spending is necessary to encourage entrepreneurial activity and innovation, which, in turn, can drive per capita output (Meierrieks 2014; Feldman et al. 2012). In contrast, others maintain that while R&D expenditure is significant, it alone cannot drive growth, and other factors, such as regulatory environment, access to capital, and market conditions, also play a crucial role (Brem and Wolfram 2014; Pece et al. 2015). This study aims to investigate the link between R&D expenditure, per capita output, and entrepreneurship, with a particular emphasis on determining how R&D expenditure influences these factors and the situations in which it is most effective in promoting growth and entrepreneurial activity.

According to Feldman (2014), innovation and entrepreneurship are essential contributors to per capita output, employment generation, and increased influence in numerous nations globally. However, promoting an innovative and entrepreneurial culture is a challenging task that requires significant R&D investments to generate discoveries, products, and markets (Brem and Wolfram 2014). R&D expenditure plays a crucial role in encouraging technological innovation and driving per capita output. The benefits of R&D investments include the creation of new employment opportunities, increased productivity, and long-term per capita output through developing novel products and services (Freeman 2018). In this context, R&D spending can be a significant driver of entrepreneurship, as it allows entrepreneurs to establish new businesses and markets based on innovative technologies and products.

Numerous research studies, such as those conducted by Feldman et al. (2016), Meierrieks (2014), and Zafar et al. (2019), have established a connection between R&D expenditure and per capita output. The OECD's research findings indicate a positive correlation between R&D expenditure and economic development across countries (Nair et al. 2020). The study further showed that R&D expenditure has a long-term impact on per capita output, with the effect being especially strong in high-income nations. Similarly, several research studies have demonstrated that R&D funding promotes entrepreneurship. Goldschlag and Miranda (2020) discovered that increased R&D expenditure leads to a rise in entrepreneurship, particularly in the high-tech sector. Foster et al.'s (2020) research concluded that firms that invest in R&D are more likely to introduce new products, and the introduction of new products is associated with higher levels of entrepreneurship.

Developing countries that produce oil, including those in the Gulf Cooperation Council (GCC), have witnessed significant per capita output due to the spike in oil prices. This has led to a rise in individual wealth and improved human development in the GCC countries (Al-Abbas 2012). However, the current decline in the oil industry and income regression, when compared to the rapid growth of other Asian nations, has compelled the GCC nations to reassess their development model. For decades, their development had relied heavily on oil revenues, and diversification of income sources is now necessary to achieve sustainable development for future generations (Baporikar 2015). Consequently, there is a greater need for entrepreneurial activity in these countries than ever before.

The connection between R&D spending, economic development, and entrepreneurship is generally considered positive, but there are some limitations to it. Bozkurt (2015) found that the connection between R&D spending and economic development is not straightforward, and the returns on investment in R&D decrease as the amount of investment increases. This indicates that the benefits of R&D investment may be declining and that other factors, such as access to funding and a favourable regulatory framework, are required to promote entrepreneurial activity and per capita output (Kritikos 2014).

Despite the established connection between R&D spending, per capita output, and entrepreneurship, numerous challenges need to be resolved before R&D investment can have a substantial impact on promoting per capita output and driving entrepreneurial activity (Brown et al. 2012). Among the most pressing issues is the increasing cost of R&D spending. R&D investment necessitates significant financial resources, which can pose a significant challenge for small and medium-sized enterprises (SMEs) and startups (Brown

et al. 2012). Additionally, the high cost of R&D investment might result in a concentration of R&D investment in a few large corporations, thereby limiting the potential for innovation and entrepreneurship. Furthermore, a lack of access to capital is a critical issue for SMEs and startups. Without access to financing, these businesses may find it difficult to invest in R&D, develop new products and services, and expand into new markets (Testa et al. 2019). This lack of access to capital can be aggravated by a weak regulatory environment that does not provide sufficient support for entrepreneurial activities.

The COVID-19 epidemic has had a tremendous influence on research and development investment and entrepreneurial activity. The epidemic has disrupted global supply lines, reduced consumer demand, and raised economic instability (Hobbs 2021). These factors have resulted in a decrease in R&D spending, limiting the potential for innovation and entrepreneurship (Galindo-Martín et al. 2021). Furthermore, the pandemic has emphasised the need for increased R&D investment to meet growing concerns, such as the need for new vaccinations and treatments for infectious diseases, as well as the development of new technology to support remote work and online learning.

A multifaceted and intricate connection exists between RED, economic expansion, and entrepreneurship. The precise mechanisms through which R&D expenditure impacts entrepreneurship and per capita output are still up for debate, despite the general agreement that R&D investment is essential to driving innovation and per capita output. As a result, the goal of this research paper is to investigate this connection in greater detail, with a particular focus on determining the circumstances under which R&D expenditures are most effective at propelling entrepreneurial activity and economic expansion. The current research is significant because it may offer policymakers useful insights. For effective policies that encourage per capita output, it is essential to comprehend the factors that drive growth. Policymakers can use this research to figure out how entrepreneurship and R&D contribute to per capita output and how they can be used to spur innovation and development. Policymakers can design policies that encourage entrepreneurial activity, increase investment in R&D, and promote per capita output, resulting in a more prosperous and sustainable future, through determining the relationship between R&D expenditure, per capita output, and entrepreneurship.

The literature on the relationship between entrepreneurship and per capita output has shown mixed results, with some studies indicating a positive relationship and others suggesting no significant relationship. Additionally, the role of research and development (R&D) support as a moderator in this relationship has not been extensively studied, particularly in the context of the Gulf Cooperation Council (GCC) countries. Therefore, this study aims to fill this gap through examining the moderating role of R&D support in the relationship between entrepreneurship and per capita output in the GCC countries. The originality of this study lies in its focus on the GCC countries, which are characterized by unique cultural, economic, and institutional factors that may influence the relationship between entrepreneurship and per capita output. Furthermore, this study examines the moderating role of R&D support, which has received limited attention in previous studies on this topic.

The paper is structured as follows: the next section provides a literature review on the relationship between entrepreneurship and per capita output as well as the moderating role of R&D support. The following section describes the research methodology, including the data collection and analysis methods. The results of the analysis are presented and discussed in the subsequent section. Finally, the paper concludes with a summary of the findings, implications, and suggestions for future research (Daunfeldt et al. 2016).

## 2. Literature Review and Hypothesis Development

### 2.1. Overview of Entrepreneurship

Entrepreneurship is the procedure of establishing, organising, and operating a new trade to obtain profits and also taking the financial risk associated with the business (Picken 2017). Entrepreneurship is gaining traction in the GCC countries, which include Bahrain,

Kuwait, Oman, Qatar, Saudi Arabia, and the United Arab Emirates. These countries are known for their oil-based economies; however, they are now diversifying their economies, and entrepreneurship is the key driver for growth and job creation (Miniaoui and Schilirò 2016). In recent years, governments have launched initiatives and programs to support entrepreneurship. For instance, the United Arab Emirates (UAE) launched the Dubai SME 100 program, which aims to identify and promote the top 100 SMEs in the region. The UAE has also launched the Mohammed bin Rashid Establishment for SME Development, which provides various forms of support for entrepreneurs, including training, mentoring, and funding (Okasha 2020).

The Qatar Business Incubation Centre, which supports start-ups across numerous industries, was also established in Qatar. The emergence of female entrepreneurs is another important development in the GCC region. Women have historically faced difficulties starting and running businesses in the GCC nations because of cultural and legal constraints. Governments have started programmes to aid female business owners. For instance, the Women's Business Council in Saudi Arabia seeks to advance gender equality in the workplace and raise the proportion of women in leadership roles (Alotaibi et al. 2017). The Dubai Business Women Council offers networking opportunities and support for female entrepreneurs in the UAE. The region's emphasis on technology firms is another trend. The young, tech-savvy populations of the GCC nations offer chances for businesses to create novel solutions. Governments in the region have started programmes to help tech companies, such as the Dubai Future Accelerators programme, which links start-ups with public sector organisations to create creative solutions (Aminova et al. 2020).

### 2.2. Relationship between Entrepreneurship and per Capita Output

Entrepreneurship plays a noteworthy part in promoting economic development. New entrepreneurial businesses contribute to the expansion of the economy through producing new occupations, aggregating the production of goods and services, and generating new revenue streams (Bjørnskov and Foss 2016). Entrepreneurs create new markets and increase competition, stimulating innovation, creativity, and efficiency. They bring new notions and technologies to the market, leading to improvements in products and services. Hence, they increase consumer demand, which drives per capita output. Moreover, the study of Teixeira and Queirós (2016) elucidates that entrepreneurs also contribute to the expansion of human capital along with the creation of new jobs and generating new revenue streams. They provide opportunities for people to acquire new skills, knowledge, and experience from their businesses. These skills are transferable, which means that they can be used in other industries or businesses, further contributing to per capita output. Furthermore, the research of Mosteanu (2019) demonstrated that entrepreneurship attracts foreign investment, and GCC countries are already attractive destinations for foreign investors due to their strategic location, natural resources, and economic stability. Entrepreneurship adds to this attraction, as start-ups and small and medium-sized enterprises (SMEs) can offer new and unique investment opportunities for foreign investors. Such investments, in turn, subsidize the formation of jobs and stimulate the development of the local economy.

Bosma et al. (2018) evaluated the association between entrepreneurship, institutions, and economic development, particularly taking evidence from Europe. The researchers collected data from 25 of the present 28 European Union countries while incorporating the available annual data from 2003 to 2014. In particular, the study's conclusions about entrepreneurship and economic development show that productive entrepreneurship supports economic progression. Furthermore, Saberi and Hamdan (2019) examined the association between entrepreneurship and economic development considering the moderating role of governmental support, particularly in the GCC countries. The researcher collected data from six GCC nations in a time series format including the 10 years from 2006 to 2015. The findings of the study stated that governmental support created a noteworthy moderating influence on the association between entrepreneurship and economic development. The study also stated that robust entrepreneurial investment indicators are

found to be high-growth and -risk capital that shows rapid development in the investment of entrepreneurial activities, while the indicators that score the lowest are found to be innovation process and technology absorption.

In contrast, the study of Savrul (2017) evaluates the effect of entrepreneurship on economic evolution. The research used data from 35 nations considering the time period of period 2006 to 2015. The data were gathered from the Global Entrepreneurship Research Association databases and the OCED. The findings of the study stated that modifications in the entrepreneurial variables and activities did not create an immediate influence on economic development. However, they influenced economic development in the long run. Based on the results, the study further suggested that the government is required to make policies concerning entrepreneurship on a long-term basis.

### 2.3. R&D Support for Entrepreneurial Investments and per Capita Output

R&D (Research and Development) support is crucial for entrepreneurial investment as it enables companies to develop innovative products and services, increase productivity, and gain a competitive edge. This support can take various forms, including grants, tax incentives, and access to funding. One of the most significant benefits of R&D support for entrepreneurial investment is that it allows startups and small businesses to conduct research and experiment with new ideas without having to worry about the high costs associated with these activities. In many cases, R&D is prohibitively expensive, making it difficult for smaller businesses to capitalise on it. With R&D support, entrepreneurs have access to the resources they need to develop and test new products, services, and technologies (Gimenez-Fernandez et al. 2020; Dellink et al. 2017). In addition, a study conducted by Wang (2018) posited that governments and other organizations play an essential role in providing R&D support to entrepreneurs. It will help drive innovation, boost per capita output, and create new opportunities for businesses to succeed through creating policies and programs that encourage R&D investment.

The link between entrepreneurship and per capita output is significantly influenced by support for research and development (R&D). The connection between entrepreneurship and per capita output is intricate and multifaceted, and it is impacted by several different circumstances (Bozkurt 2015). According to Castaño et al. (2016), one such element that influences the link between entrepreneurship and per capita output favourably is R&D assistance. R&D support reassures entrepreneurs to produce innovative goods and services that drive per capita output. The above-mentioned study also suggests that governments are required to raise expenditures on education and research and development. It encourages a culture of entrepreneurship, decreases complex administration, and increases the financial sustenance of SMEs, particularly for entrepreneurs, as they hold the capability to create jobs. Additionally, R&D support aids entrepreneurs in overcoming the upfront expenses and risks connected with producing new goods and services through offering financing and resources for research and development. It leads to the formation of new industries and the growth of prevailing ones, which in turn drive per capita output (Foster et al. 2020; Lindholm-Dahlstrand et al. 2019).

Furthermore, Pece et al. (2015) explained in the study that investments in technology and expenditure related to R&D and innovation are the premises for ensuring progress and competitiveness and directing them toward sustainable economic development. Economic development is endogenously defined and impacted by the decisions of agents to enhance profits and give consideration to factors linked with entrepreneurial activities, demonstrating the innovation procedure based on microeconomic information. R&D support also helps businesses and entrepreneurs to remain competitive in the global marketplace. R&D support helps companies create goods and services that are competitive and compliant with international standards through giving them access to new technology and information. This enables businesses to expand their market share and generate more revenue, which contributes to per capita output (Lüdeke-Freund 2020).

Regardless of the well-established connection between entrepreneurship, per capita output, and R&D spending, there are various problems that need to be resolved completely to evaluate the effectiveness of R&D investment in raising entrepreneurial activity and driving economic progress. Lack of financial access for SMEs and start-ups is one of the major problems. These businesses may find it difficult to capitalise on R&D, create new goods and services, and penetrate new markets if they lack access to financing. In addition, the rising cost of R&D expenditure is another important issue as for small and medium-sized businesses (SMEs) and startups, the requirement for large financial resources for R&D investment can be a major hurdle (Testa et al. 2019).

*2.4. Theoretical Framework*

2.4.1. Innovation Theory

An innovation signifies an object or an idea which is observed to be new. Innovation theory states that the diffusion rate is influenced by the considerable advantage of complexity, trial-ability, innovation, compatibility, and observability (Keklik 2018). Regarding entrepreneurship and per capita output, innovation theory suggests that entrepreneurship and per capita output are interrelated and that entrepreneurial events have a critical role in driving per capita output. Innovation theory identifies entrepreneurship as the primary source of innovation, which, in turn, is the engine of economic development (Urbano et al. 2019; Silaghi et al. 2014). Innovation theory suggests that entrepreneurship contributes to per capita output in several ways. First, entrepreneurs form new goods and services which meet emerging needs in the market. This innovation increases the efficiency and productivity of the economy, resulting in higher per capita output. In addition to that, innovation theory also suggests that the atmosphere in which entrepreneurs operate plays a critical role in their ability to innovate and drive per capita output. Factors such as access to capital, a skilled workforce, supportive government policies, and a strong legal framework are all essential to creating an environment that fosters entrepreneurship and innovation (Saberi and Hamdan 2019). Scuotto et al. (2022) discuss the impact of disruptive technologies, such as artificial intelligence (AI), on small and medium-sized enterprises (SMEs). The study highlights the importance of individual technology absorptive capacity (TAC) in driving the effective and efficient use of such technologies by Chief Information Officers (CIOs) within SMEs. The authors argue that the involvement of CIOs at the micro level of the firm can help encourage technology absorptive capacity at the meso level.

2.4.2. The Theory of Economic Development

The economic development theory which was proposed by Joseph A. Schumpeter states that entrepreneurial activities cause economic development through enabling the production means in a community to be utilized in an innovative and more proficient combination. Schumpeter argues that technological innovation is driven by entrepreneurship rather than just knowledge. After that, he makes the case that entrepreneurship is a strategy, not a consequence, of a person's rational economic activity. He argues that in order for individuals to act in an economically reasonable way, they must have some information upon which to make their choices (Smith 2010; Zolas et al. 2021). The study of Saberi and Hamdan (2019) states that the theory of economic development recognizes that entrepreneurship is essential for starting new enterprises, creating novel goods and services, and expanding into untapped areas. Entrepreneurship can take many forms, from small businesses to large corporations, but all types of entrepreneurship share the common goal of creating value and generating wealth. This wealth creation is a key driver of per capita output, as it increases the overall size of the economy and generates new opportunities for investment, trade, and employment.

*2.5. Conceptual Framework and Hypothesis Development*

The discussion presented above clearly demonstrates the impact of entrepreneurship on per capita output. Furthermore, it reveals that the relationship between entrepreneurship and per capita output is influenced by R&D support, as depicted in Figure 1: Study Model.

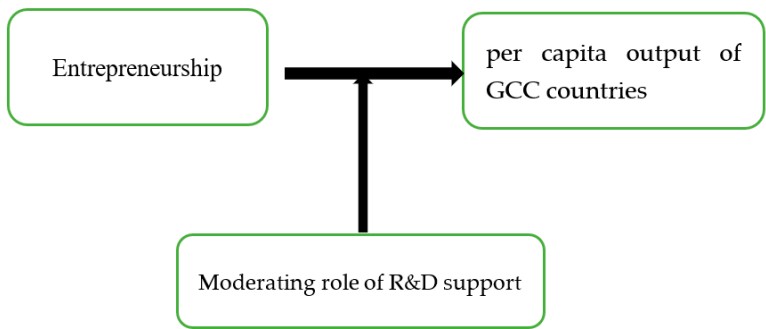

**Figure 1.** Study Model.

**H1.** *Entrepreneurship has a significant effect on per capita output.*

**H2.** *The relationship between entrepreneurship and per capita output is moderated by R&D support.*

## 3. Methodology

*3.1. Data Sources*

The data sources for the current study were covered through the examination of six GCC countries: Saudi Arabia, Bahrain, Kuwait, Qatar, and Oman. For statistical analysis, data spanning 10 years from 2010 to 2020 were obtained for each of the six countries, which included their expenditures and economic indicators. Data for these indicators were collected from a myriad of authentic sources, of which two major ones were the World Bank Database and GCC online portals.

*3.2. Measurement*

To examine the moderating effect of R&D expenditure, this study employs R&D expenditure as a percentage of GDP as an indicator. This metric is commonly used in academic literature to gauge a country's R&D investment level. R&D investment is an essential driver of innovation and technological progress, which can create novel products and services, improve productivity, and stimulate long-term economic expansion. Moreover, R&D investment can play a critical role in entrepreneurship through enabling business owners to establish new businesses and markets based on cutting-edge technologies and products. This study employs per capita output as the dependent variable, as it is a vital metric of a country's development level. Additionally, per capita output significantly impacts a population's quality of life through influencing employment opportunities, income levels, and social welfare.

Two indicators were used in the study to measure entrepreneurship: the scores of Ease of Doing Business and the Global Entrepreneurship Index (GEI). The GEI is a composite index that measures the level of start-up activity, the availability of funding, and the support provided via government policies in a country's entrepreneurial ecosystem (Ali et al. 2021). In addition, Acs et al. (2014) say that the Ease of Doing Business score measures how easy it is to start and run a business in a country, such as how easy it is to get permits, how efficient the regulatory environment is, and how easy it is to get credit. A comprehensive picture of a nation's entrepreneurial activity is provided by these indicators. Ease of Doing Business scores provide a more in-depth picture of the regulatory and financial environment for entrepreneurship, while the GEI is particularly useful for measuring the quality of the entrepreneurial ecosystem. In addition, in the context of GCC nations, the selection of the

GEI and Ease of Doing Business scores as indicators of entrepreneurship is particularly pertinent (Acs et al. 2014; Witt 2016). In an effort to diversify their economies away from oil dependence and encourage entrepreneurship, these nations have launched a number of initiatives. As a result, this study's use of the GEI and Ease of Doing Business scores is particularly appropriate for assessing the level of entrepreneurship in GCC nations and determining the factors driving its expansion.

The moderator variable was considered a second independent variable since it adjusts the degree and course of the relationship between the dependent and independent variables. For a variable to be assigned a moderator, it should have a causal connection between the reliant variable (financial development) and the arbitrator variable (government support). Moreover, the independent variable ought to be irrelevant to the moderator. To show and survey the impact of the moderator variable on the connection between entrepreneurship and economic development, the following methodologies were utilised.

In the first step, the influence of entrepreneurship (Ease of Doing Business + Global Entrepreneurship Index) was investigated in this model with other fundamental variables (labour force and capital accumulation) without taking R&D assistance into account:

$$\text{GDP}_{it} = \alpha + \beta_1 \, Eseb_{it} + \beta_2 \, \text{Enterp}_{it} + \beta_1 \, Lab_{it} + \beta_2 \, Capit_{it} + \varepsilon_t$$

Afterwards, in addition to the interaction variables (research and development expenditure (% of GDP) between entrepreneurship and government support: entrepreneurship in the GCC), a dependent variable referring to entrepreneurship in the GCC was included.

$$\text{GDP}_{it} = \alpha + \beta_1 \, Eseb_{it}(\, Rde_{it}) + \beta_2 \, \text{Enterp}_{it}(\, Rde_{it}) + \beta_1 \, Lab_{it} + \beta_2 \, Capit_{it} + \varepsilon_t$$

## 4. Results

### 4.1. Entrepreneurship in Gulf Cooperation Council

Table 1 presents entrepreneurship ranking, scoring, and indicators of GCC countries for the year 2019. Notably, the UAE stands at the top rank amongst the six countries in terms of the Global Entrepreneurship Index and ranks at 25. Qatar comes next, ranking at 28; then Bahrain, ranking 38; at fourth, Oman, ranking 39; and then followed by Saudi Arabia and Kuwait. On the contrary, in terms of entrepreneurial attitudes, Saudi Arabia comes first ranking 21, UAE comes next at a rank of and then followed by Qatar, Bahrain, Oman, and Kuwait onwards. Table 1 also refers to indicators of entrepreneurship in GCC countries. Referring to opportunity perception and start-up skills, Saudi Arabia ranks highest and scores 0.717 and 0.981, respectively. As for risk acceptance, Kuwait has the highest score (i.e., 0.491), and Bahrain and Oman have the lowest (i.e., 0.228). In terms of networking, both Saudi Arabia and the UAE have the highest and are recorded as 1. Additionally, the UAE also has the highest cultural support, and it has been recorded as 1, while the lowest cultural support score is found in Bahrain and is recorded as 0.281.

Further, when referring to start-up opportunity, Qatar has the highest score (i.e., 0.714), followed by the UAE, Bahrain, Kuwait, and Saudi Arabia, onwards. Technological absorption is another major indicator of entrepreneurship, and Bahrain has the highest score, whilst Saudi Arabia has the lowest score. In human capital, most of the GCC countries have a high score, with the UAE scoring the highest (i.e., 0.942), followed by Bahrain, Qatar, Oman, Kuwait, and Saudi Arabia, respectively. In terms of competition and product innovation, Bahrain has the highest score. Moreover, in process innovation, the UAE has the highest record at 0.601. In terms of growth, Saudi Arabia has recorded the highest score, closer to 1. Lastly, in terms of risk capital, Bahrain has the highest risk capital, and it has been recorded as 0.91. Thus, findings revealed that GCC countries have achieved the highest score in three indicators of entrepreneurship, including human capital, high growth, and risk capital.

**Table 1.** Entrepreneurship in the Gulf Cooperation Council.

| Entrepreneurship in GCC | Saudi Arabia | Bahrain | Qatar | Oman | UAE | Kuwait |
|---|---|---|---|---|---|---|
| *Ranking* | | | | | | |
| Global Entrepreneurship Index, Rank | 42 | 38 | 28 | 39 | 25 | 47 |
| Entrepreneurial Attitudes, Rank | 21 | 44 | 33 | 52 | 22 | 61 |
| *Score* | | | | | | |
| Entrepreneurial Attitudes, Score | 56.8 | 36.51 | 43.03 | 35.18 | 56.63 | 33.26 |
| Global Entrepreneurship Index, Score | 42.1 | 43.8 | 51.6 | 43.6 | 54.2 | 37.4 |
| *Indicators* | | | | | | |
| Opportunity perception | 0.717 | 0.603 | 0.587 | 0.448 | 0.529 | 0.439 |
| Start-up skills | 0.981 | 0.468 | 0.166 | 0.341 | 0.55 | 0.165 |
| Risk Acceptance | 0.468 | 0.228 | 0.347 | 0.228 | 0.303 | 0.491 |
| Networking | 1 | 0.495 | 0.674 | 0.409 | 1 | 0.454 |
| Cultural Support | 0.476 | 0.281 | 0.701 | 0.39 | 1 | 0.32 |
| Opportunity start-up | 0.472 | 0.632 | 0.714 | 0.61 | 0.661 | 0.603 |
| Technology Absorption | 0.154 | 0.364 | 0.38 | 0.341 | 0.233 | 0.325 |
| Human capital | 0.344 | 0.904 | 0.859 | 0.635 | 0.942 | 0.626 |
| Competition | 0.307 | 0.449 | 0.547 | 0.254 | 0.563 | 0.226 |
| Product Innovation | 0.343 | 0.564 | 0.837 | 0.485 | 0.622 | 0.422 |
| Process Innovation | 0.136 | 0.108 | 0.548 | 0.284 | 0.601 | 0.313 |
| High Growth | 0.329 | 1 | 1 | 1 | 0.952 | 1 |
| Internationalization | 0.808 | 0.493 | 0.576 | 0.468 | 0.339 | 0.095 |
| Risk Capital | 0.729 | 0.91 | 0.731 | 0.899 | 0.586 | 0.802 |

Source: Adapted from (World Development Indicators (WDI) 2022), with permission from publisher Knoema, 2022.

### 4.2. Descriptive Statistics

Table 2 presents descriptive statistics of the variables (i.e., global entrepreneurship, ease of doing business, R&D expenditure, labour force, capital force, and GDP growth) involved in this research. It can be seen that the mean value of the Global Entrepreneurship Index is found to be 48.8, which implies the average score of global entrepreneurs in GCC countries is found to be 48.8 during the period from 2010 to 2020. Additionally, its standard deviation value is found to be 7, which implies data are more spread out in GCC countries. It also shows the minimum and maximum scores of GCC countries during the period. Notably, the maximum GEI score is found to be 63.4, and the minimum score is 37.4 in GCC countries. The average score of Ease of Doing Business is computed to be 64.2 in GCC countries, and it has been expected to deviate with a higher value; the standard deviation value of the Ease of Doing Business is 9.9. Further, referring to the maximum value, it has been estimated to be 81.6, and the minimum value is 32.2. Further, while referring to R&D, the mean value is found to be 0.4, which implies that the average expenditure of R&D in GCC countries from 2010 to 2020 is found to be 0.4%. Its standard deviation value is found to be 0.4, which implies data are less clustered and closer to the mean value in GCC countries. The maximum value of R&D expenditure in GCC countries is found to be 1.4% and the minimum is 0.0%.

Moreover, labour force and capital have also been considered as supporting variables. Notably, the average number of the labour force in GCC is 4,349,235, as the mean value is recorded as 4,349,235. However, it has been expected to deviate towards 4,368,226. The maximum number of the labour force in GCC countries is found to be 16,240,076.0 and the minimum number of the labour force is 671,597 in GCC countries. While referring to the capital force, it can be observed that GCC countries are capital-extensive as compared to labour, as the mean value of the capital force is significantly higher than the labour force and recorded as USD 221 billion. However, it has been expected to deviate with higher numbers, as the standard deviation value is found to be 274 billion. This variation in capital force is caused by variations in the size of the economies among GCC countries. The maximum value of the capital labour force is found to be USD 878 billion, and the minimum value the of capital labour force is USD 2420 million. Lastly, GDP growth has

also been considered as a measure of GDP per capita. It can be seen that the average GDP per capita in GCC countries is USD 82,116.60, and it has been expected to deviate from USD 6594, as the standard deviation value is found to be 6594. The maximum value of GDP per capita is found to be USD 261,469.80, and the minimum value of GDP per capita is USD 6594.

**Table 2.** Descriptive Statistics.

| Variables | Global Entrepreneurship Score | Ease of Doing Business | R&D (% of GDP) | Labour Force | Capital Force | GDP per Capital |
|---|---|---|---|---|---|---|
| Mean | 48.8 | 64.2 | 0.4 | 4,349,235.0 | 221,000,000,000.0 | 82,116.6 |
| Median | 46.3 | 65.3 | 0.2 | 2,253,156.0 | 76,800,000,000.0 | 41,890.4 |
| Maximum | 63.4 | 81.6 | 1.4 | 16,240,076.0 | 878,000,000,000.0 | 261,469.8 |
| Minimum | 37.4 | 32.2 | 0.0 | 671,597.0 | 2,420,000,000.0 | 6594.0 |
| Std. Dev. | 7.0 | 9.9 | 0.4 | 4,368,226.0 | 274,000,000,000.0 | 89,466.2 |
| Skewness | 0.6 | −0.8 | 1.2 | 1.4 | 1.1 | 0.9 |
| Kurtosis | 2.2 | 4.0 | 3.8 | 3.6 | 3.0 | 2.3 |
| Observations | 66 | 66 | 66 | 66 | 66 | 66 |

*4.3. Panel Regression Model*

A panel regression (i.e., fixed or random effect) model has been used due to cross-section and time series data. For differentiation between fixed effects and random effects, the Hausman test is used, in which the null hypothesis assumes that the random effect model is applicable (Ghosh and Sanyal 2019). Meanwhile, the alternative hypothesis indicated that only the fixed-effect model can be used for analysis. However, from Table 3 below, both models are statistically significant, as P values are found to be less than 0.05, which means that the fixed-effect model (FE) is used to ascertain the relationship between entrepreneurship and per capita output and the moderating role of R&D between entrepreneurship and per capita output.

**Table 3.** Panel Regression Model.

| | Model 1 Fixed Effect | | Model 2 Random Effect | |
|---|---|---|---|---|
| Variable | Coefficient | t-Statistic | Coefficient | t-Statistic |
| Constant | 97,226.48 | 34.726 *** (0.000) | 86,635.44 | 15.238 *** (0.000) |
| *Entrepreneurship* | | | | |
| Ease of doing business | −37.35 | −2.311 ** (0.024) | | |
| Global Entrepreneurship Index, score | −246.54 | −3.925 *** (0.000) | | |
| *R&D Role* | | | | |
| Ease of doing business × RDE | | | 157.892 | 0.636 (0.526) |
| Global Entrepreneurship Index × RDE | | | 27.920 | 0.077 (0.938) |
| *Supporting Variables* | | | | |
| Labour Force | −0.001 | −6.710 *** (0.000) | −0.003 | −2.402 ** (0.019) |
| Capital | 0.00 | 3.676 *** (0.000) | 0.00 | 0.840 (0.404) |
| R-squared | 0.998 | | 0.996 | |
| Adjusted R-squared | 0.998 | | 0.996 | |
| F-statistic | 5876.63 | | 1704.118 | |
| Prob(F-statistic) | 0.000 | | 0.000 | |
| Durbin–Watson stat | 1.102727 | | 0.459 | |
| *Hausman test* | | | | |
| Chi-square statistic | 4 *** | | 15.911 *** | |
| Prob. (Chi-square) | 0.000 | | 0.003 | |

Note: ** indicate significance at 5%, and *** indicate significance at 1% levels, respectively.

From the above table, Model 1 reflects the impact of entrepreneurship (including Ease of Doing Business and Global Entrepreneurship Index) on per capita output (GDP

per capita) in GCC countries. Ease of Doing Business has a negative and significant influence over GDP growth, as coefficient values are found to be −37.35, and its sig value is 0.024 < 0.05. Similarly, global entrepreneurship has also a negative and significant influence over GDP per capita, as the coefficient value is −246.54, and the sig value is 0.000 < 0.01. Similarly, findings in previous studies have also found that modifications in the perineurial activities and variables do have not any immediate influence on economic development and growth (Savrul 2017). Thus, it is suggested that entrepreneurship has a negative influence on per capita output in GCC countries. Moreover, labour force and capital force have also been considered in this model as supporting variables. It can be observed that the labour force has a negative and significant influence over GDP per capita, as the coefficient value is determined to be −0.001 and the sig value is 0.000 < 0.01. This negative influence of labour is due to increasing labours that tend to decrease the GDP per capita of GCC countries. On the contrary, the capital force has not any influence over GDP per capita, as the coefficient value is found to be 0.000, and it is significant at a 1% level.

Furthermore, the moderating variable R&D expenditure (% of GDP) has been considered to analyse its support in the relationship between entrepreneurship and per capita output in Model 2. It can be seen that Ease of Doing Business moderating with RDE has a positive but insignificant influence over GDP growth, the as coefficient value is found to be 157.8 and the sig value is 0.526 > 0.1. Similarly, Global Entrepreneurship Index moderating with RDE has a positive but insignificant influence over GDP growth, as the coefficient value is found to be 27.9 and the sig value is 0.077 > 0.1. Thus, it implies that supporting RDE has no significant influence on the relationship between R&D and per capita output. Further, Model 2 also shows that the labour force has a negative and significant influence over GDP per capita, as the coefficient was determined to be −0.003 and the sig value is 0.019 < 0.05. On the contrary, the capital force has not any influence over GDP per capita, as the coefficient value is found to be 0.000, but it is found to be insignificant as the sig value is found to be 0.404 > 0.1. Further, while referring to the above table, it is manifest that the adjusted R2 was higher in Model 1 than in Model 2. It revealed that the interactive variable (research and development expenditure) has not had any statistical significance for the per capita output of GCC countries.

## 5. Discussion and Hypothesis Testing

The main emphasis of this research was to analyse the role of entrepreneurship in the per capita output of GCC countries (including Saudi Arabia, the UAE, Qatar, Bahrain, Oman, and Kuwait). However, based on the findings, it has been determined that Ease of Doing Business as an indicator of entrepreneurship has a significant influence on GDP per capita. Additionally, the Global Entrepreneurship Index has also recorded a negative and significant influence on GDP per capita. Thus, it shows that an increase in entrepreneurship tends to decrease GDP per capita. Similarly, findings in previous studies have also found that modifications in entrepreneurial activities and variables have no immediate influence on economic development and growth (Savrul 2017). The author suggested that studies of government policies and interventions are required concerning entrepreneurship and per capita output. Further, Saberi and Hamdan (2019) indicated in their study that rapid development in the investment of entrepreneurial indicators in the innovation process and technology absorption is a major reason that GDP per capita tends to decrease. Furthermore, the negative impact of R&D on per capita output may be attributed to the time lag between investment in R&D and per capita output. Additionally, due to the limited knowledge base and scientific discoveries in GCC countries, achieving rapid innovation may require substantial investment in R&D. Therefore, entrepreneurs in GCC countries may opt to purchase patents, licenses, and copyrights for inventions and discoveries from other countries rather than investing in R&D domestically. This approach may lead to per capita output without significant investment in R&D. According to the findings of the present study, H1 has been confirmed and accepted, as demonstrated in Table 4.

**Table 4.** Hypothesis Testing Summary.

| | Hypothesis Statement | Accepted/Rejected |
|---|---|---|
| H1 | H1. There is an effect of entrepreneurship on per capita output. | Accepted |
| H2 | H2. There is an effect of R&D support on the relationship between entrepreneurship and per capita output. | Rejected |

It suggests that although R&D investments are crucial for per capita output, they may not have a direct impact on the relationship between entrepreneurship and per capita output. This finding could be interpreted in several ways. One possibility is that entrepreneurship and R&D may be complementary rather than substitutive, meaning that they work together to promote per capita output, but neither is more important than the other.

Alternatively, it could be that the effects of R&D investments on per capita output take longer to materialize and are more indirect, whereas entrepreneurship may have more immediate and direct effects on growth. It's also possible that the study's measure of R&D expenditures may not fully capture the complex relationship between R&D and entrepreneurship and that different measures or methodologies could yield different results.

Overall, this finding underscores the importance of examining the complex interplay between different factors that contribute to per capita output, including entrepreneurship, R&D investments, and other institutional and policy factors. Further research could help shed more light on the specific mechanisms through which these factors interact to drive per capita output, and how they may vary across different contexts and periods.

Moreover, findings in the current research on labour force and capital force have also been considered as supporting variables to analyse their influence on per capita output. Findings revealed that the labour force has a negative influence on GDP per capita in GCC countries due to an increase in the number of unskilled workers, which tends to decrease GDP per capita (Alsamara 2022). On the contrary, findings show that capital force has a positive influence over GDP per capita, which means that an increase in capital force tends to increase GDP per capita. Similarly, findings in previous studies have also revealed that access to capital, a skilled workforce, and a strong legal framework are all essential to creating an environment that fosters entrepreneurship and the highest per capita output (Teixeira and Queirós 2016; Alotaibi et al. 2017).

Further, findings in the current research have also analysed the supporting role of research and development between GDP per capita and per capita output. Findings show that RDE does not have any effect on the relationship between entrepreneurship (Ease of Doing Business and Global Entrepreneurship Index) and per capita output (GDP per capita). Similarly, some researchers have also argued that the high cost associated with R&D makes it difficult for smaller businesses to invest in it (Gimenez-Fernandez et al. 2020). Thus, based on the findings in the current research and previous studies, H2 is found to be incorrect and rejected. However, the following table provides a summary of the hypotheses tested based on the findings of the current research.

Moreover, the relationship between entrepreneurship and per capita output may be complex and depend on various contextual factors. While some studies suggest that entrepreneurship can contribute positively to per capita output, others argue that the impact may depend on the type of entrepreneurship, the stage of economic development, and the institutional environment. Therefore, it is important to interpret the regression result in light of the specific context and the limitations of the model. Let us consider the variables that were used in the regression model. The two main independent variables were "Ease of Doing Business" and "Global Entrepreneurship Index (GEI) score". Ease of Doing Business refers to the regulatory environment for starting and operating a business, while the GEI measures the quality and depth of the entrepreneurship ecosystem in a country.

The dependent variable was "per capita output," which was measured using Gross Domestic Product (GDP) per capita. The results of the regression analysis indicated that

both the Ease of Doing Business and GEI scores had a significant negative impact on per capita output. This means that in countries where it is easier to do business and where the entrepreneurship ecosystem is stronger, per capita output tends to be lower. This finding may seem counterintuitive at first glance, but there are several possible explanations. One possible explanation is that in countries where it is easier to do business and where the entrepreneurship ecosystem is stronger, there may be more competition among businesses. This increased competition may lead to lower profit margins and lower investment in new ventures, which in turn could slow down per capita output. Another possible explanation is that the Ease of Doing Business and the strength of the entrepreneurial ecosystem may be less important determinants of per capita output than other factors such as macroeconomic stability, infrastructure, and access to credit. It is important to note that the results also showed that R&D expenditures did not significantly moderate the relationship between entrepreneurship and per capita output. This means that while R&D is important for per capita output, it may not necessarily be the key factor that determines the relationship between entrepreneurship and per capita output. It is noteworthy that although previous theories postulate a positive association between expenditures on research and development (R&D) and per capita output, empirical studies do not always support this relationship. The present study provides several potential reasons for the absence of a moderated relationship. Firstly, the sample size may not be representative of all countries, or the data may not fully reflect R&D expenditures in each country. Secondly, the measure employed to gauge R&D expenditures may lack the sensitivity to capture the complete impact of R&D on per capita output. Furthermore, unaccounted factors may have influenced the correlation between entrepreneurship and per capita output.

It is imperative to recognize that empirical research often reveals mixed or unexpected results, which can contribute to ongoing deliberations in the field. Consequently, further research can employ these findings as a foundation and investigate possible justifications for the lack of a moderated relationship between entrepreneurship and per capita output within the framework of R&D expenditures. It is important to note that while past theory suggests a positive relationship between R&D expenditures and per capita output, this is not always the case in empirical studies. There may be several reasons for the lack of a moderated relationship recorded in this study. One possible explanation could be that the sample used in the study was not representative of all countries, or that the data used did not capture the full extent of R&D expenditures in each country. Another possible limitation of the study is that the measure used for R&D expenditures may not have been sensitive enough to capture the full effect of R&D on per capita output. Additionally, it is possible that other factors not included in the study may have influenced the relationship between entrepreneurship and per capita output. It is important to note that empirical research often yields mixed or unexpected results, and these findings can contribute to ongoing debates and discussions in the field. Further research can build on these findings and explore potential explanations for the lack of a moderated relationship between entrepreneurship and per capita output in the context of R&D expenditures.

## 6. Conclusions

The current study was conducted to decipher the moderating role of R&D expenditure of the government in promoting per capita output in GCC countries. In this regard, the study analysed the direct relationship between entrepreneurship and per capita output and then examined the causal impact of R&D investments. Through a thorough statistical analysis of data from 2010 to 2020, the study found that the UAE and Qatar rank higher in the Global Entrepreneurship Index than other GCC countries, indicating a higher level of entrepreneurship in these countries. Additionally, entrepreneurial attitudes and aspirations in the UAE, Bahrain, and Saudi Arabia are relatively high. These indicators suggest that entrepreneurship has the potential to contribute significantly to the per capita output of these countries. Moreover, the data also highlight the importance of R&D expenditure in

promoting entrepreneurship and per capita output. Countries with higher R&D expenditures, such as the UAE, Bahrain, and Oman, tend to have higher scores in entrepreneurial abilities, indicating a positive relationship between R&D spending and entrepreneurship. This suggests that investing in R&D can create new opportunities for entrepreneurship and promote per capita output in the long term.

The results indicate that the Ease of Doing Business and Global Entrepreneurship Index scores have a significant negative effect on per capita output in both models, suggesting that improving the business environment and entrepreneurship ecosystem can lead to higher per capita output. The coefficients of the interaction terms between Ease of Doing Business and R&D, and Global Entrepreneurship Index and R&D, are not statistically significant, indicating that R&D expenditures do not have a significant moderating effect on the relationship between entrepreneurship and per capita output. Furthermore, the supporting variables of labour force and capital have a significant positive effect on per capita output in both models, indicating their importance for per capita output. The adjusted R-squared values for both models are high, indicating that the variables included in the models explain a large proportion of the variation in per capita output.

Overall, the results suggest that entrepreneurship plays an important role in promoting per capita output, and improving the business environment and entrepreneurship ecosystem can lead to higher per capita output. R&D expenditures, however, do not seem to have a significant moderating effect on this relationship.

## 7. Future Implications of Research

This research provides important insights into the factors that influence per capita output, particularly the role of entrepreneurship and R&D expenditures. The findings of this study have important implications for academia and policy-making. In academia, this study highlights the need to further investigate the relationship between entrepreneurship and per capita output. Future research should examine the specific mechanisms through which entrepreneurship promotes per capita output as well as the factors that influence entrepreneurial activity. Moreover, this study provides a framework for future research that seeks to understand the role of R&D expenditures in driving per capita output.

In terms of policy-making, this study provides valuable information to policymakers on how to promote per capita output. The findings suggest that creating a supportive environment for entrepreneurship, as well as increasing R&D expenditures, can have a positive impact on per capita output. Therefore, policymakers should focus on creating an enabling environment for entrepreneurs, as well as increasing funding for R&D. This could include policies that reduce the regulatory burden on businesses, provide tax incentives for R&D, and encourage public–private partnerships to promote innovation.

**Author Contributions:** Conceptualization, H.B. and N.F.S.A.; methodology, S.S.; software, H.B.; validation, H.B., S.M.M.M. and B.M.A.A.; formal analysis, H.B.; investigation, N.F.S.A.; resources, H.B.; data curation, H.B.; writing—original draft preparation, I.A.E.E.; writing—review and editing, I.A.E.E.; visualization, S.S.; supervision, S.M.M.M.; project administration, B.M.A.A.; funding acquisition, H.B. All authors have read and agreed to the published version of the manuscript.

**Funding:** This research received no external funding.

**Institutional Review Board Statement:** Not applicable.

**Informed Consent Statement:** Not applicable.

**Data Availability Statement:** Not applicable.

**Conflicts of Interest:** The authors declare no conflict of interest.

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
