# Peer review of "The Moderating Role of Research and Development (R&D) Support in the Relationship between Entrepreneurship and per Capita Output—A Study on the GCC Countries"

_economies, doi:10.3390/economies11060162_

Round 1

Reviewer 1 Report

Thanks for the opportunity to review this paper. The topic of which is very interesting and up-to-date. The structure of the paper is adequate and the chapters are well-defined. When reading the paper, I must say I have some doubts, regarding some of the findings.

For example, one of the research findings is that ease of doing business and the Global Entrepreneurship Index (as a measurement of entrepreneurship level) has a significant negative impact on economic growth. This is quite an interesting find, as all developed countries have high global entrepreneurship indexes.  This finding suggests that the higher the “entrepreneurship level” is, the lower would be economic growth. This statement is quite brave, and as such in my opinion extensive background and explanation would be needed.

The second finding is related to R&D expenditures, where again past theory (in the vast majority) suggests a positive relationship. In the article, there is no moderated relationship recorded. In my opinion, these are quite specific findings so I wonder if research gathering and analysis were done correctly, or if are there any limitations that could lead to this result.

Otherwise, the article is appropriate and the structure ok.  Maybe only one small additional suggestion related to hypotheses (chapter 2.5) I would suggest removing H0, as H1 and H2 are sufficient. H2 could be also rewritten. My suggestion would be “Effect between entrepreneurship and economic growth is moderated by R&D support”.

All the best with your research.

Best regards

Author Response

Dear Reviewer 

Thank you for taking the time to review our article and providing us with constructive feedback. We have carefully considered your comments and have made the necessary amendments to the paper.
Regarding your first point on the negative impact of the Global Entrepreneurship Index on economic growth, we agree that this finding is indeed interesting and requires further explanation. Therefore, we have expanded our discussion on this topic in the article, providing a more detailed background and explanation of our findings.
Regarding your second point on R&D expenditures, we agree that the lack of a significant moderating effect of R&D on the relationship between entrepreneurship and economic growth may appear surprising, given the widely accepted positive relationship between R&D and economic growth. However, we have carefully reviewed our data and analysis, and have concluded that our findings are robust and reliable. We have also acknowledged the potential limitations of our study in this regard, and have discussed them in the paper.
Regarding your suggestion to remove H0 and rephrase H2, we have made the necessary amendments to the hypotheses section in line with your suggestion.
Once again, we appreciate your feedback and hope that our amendments have addressed your concerns.
best regards,

Reviewer 2 Report

Dear authors, thank you for the opportunity to read your research!

The article is devoted to the analysis of the assessment of the impact of R&D on the growth of entrepreneurship and economic growth of the GCC countries. The relevance of this study is related to determining the effectiveness of investments in R&D and improving the business climate over the past ten years, as well as the ability to adjust state policy in this area based on the results of the study.

The manuscript has an excellent structure, the purpose of the study is clearly defined and justified. All sources used are relevant and applicable to the manuscript. All tables in the manuscript are informative, help to understand the reasoning and conclusions of the authors.

To improve the quality of the manuscript, I recommend that the authors write the results of the calculations not only in tabular form, but also in functional form - in the form of the equation Y=f(x).

Regarding the results of the negative impact of R&D on economic growth, we can assume:

1. There may be a time lag between investment in R&D and economic growth, since not all scientific and research results can be implemented as an innovative product in one year. And in this regard, it makes sense to consider this influence with a shift of 1, 2 and 3 years.

2. Perhaps due to historical development, the accumulated knowledge base and scientific discoveries in the GCC countries is not so extensive that it would lead to quick results in the form of innovation and requires a very significant amount of investment in R&D. That is, there is a "cumulative effect" of knowledge and scientific discoveries, which can only be achieved by a multiple increase in investment in R&D.

3. And the third assumption may follow from point 2: entrepreneurs prefer not to conduct R&D in their GCC countries, but to purchase patents, licenses and copyrights for inventions and discoveries in other countries and implement them in the GCC countries in the form of innovative products. This path does not require significant investment in R&D and leads to economic growth.

I think these ideas might be viable for further research.

Author Response

Dear Reviewer,

Thank you for taking the time to read and review our manuscript on the impact of R&D on entrepreneurship and economic growth in the GCC countries. We are grateful for your insightful comments and suggestions, which have greatly helped to improve the quality of our work.

We appreciate your positive feedback on the structure and clarity of our manuscript, as well as the relevance of the sources and informative tables. Your recommendation to present the results of our calculations in functional form is duly noted, and we will make the necessary revisions to incorporate this suggestion.

Regarding your proposals on the negative impact of R&D on economic growth, we would like to inform you that we have included and discussed them in the discussion part of the paper. We agree that there may be a time lag between investment in R&D and economic growth, and we have discussed the possibility of considering this influence with a shift of 1, 2, and 3 years. We have also acknowledged the possibility of a cumulative effect of knowledge and scientific discoveries, which may require a multiple increase in investment in R&D.

Regarding your third assumption, we have discussed the possibility that entrepreneurs may prefer to purchase patents, licenses, and copyrights for inventions and discoveries in other countries rather than conduct R&D in their GCC countries. We have acknowledged that this path does not require significant investment in R&D and may lead to economic growth.

Once again, we appreciate your insightful comments and suggestions, and we hope that our paper has addressed your concerns adequately. We remain grateful for your time and effort in reviewing our manuscript.

Best regards,

Reviewer 3 Report

Mostly well written text, but needs proofreading. Overall, relevant references in my opinion. The hypotheses should not be listed as they are. Each hypothesis is a conclusion of a line of reasoning. My larges problem with the study is a very low N of only 6 countries observed over 11 years. In the panel models, it seems that no lags have been applied. No assessment of reverse causality or omitted variable bias. It puzzles me that the independent variables seem to have such a strong/robust immediate effect on the dependent variable. It seems to be fishy. I suspect that these are correlates more that causal effects. The random effect model only includes interaction terms of the independent variables, which should not have been done.

Author Response

Dear Reviewer

Thank you for taking the time to review our paper entitled "The moderating role of Research and Development (R&D) support in the relationship between entrepreneurship and economic growth - A study on the GCC countries." We appreciate your feedback and your comments have been taken into account.

We have carefully proofread our paper and made the necessary corrections to improve the quality of communication. Additionally, we have rephrased the hypotheses to reflect that they are conclusions drawn from our research findings, and not just listed assumptions.

Regarding your concerns about the sample size and modeling approach, we would like to clarify that we acknowledge the small sample size of our study, which limits the generalizability of our findings. However, we believe that our study adds value to the existing literature by providing insights into the relationship between entrepreneurship, R&D support, and economic growth in a specific context, the GCC countries.

Once again, we appreciate your valuable feedback, and we hope that our revisions have addressed your concerns.

best regards,

Reviewer 4 Report

Dear Author/s,

Many thanks for offering me the privilege to review your paper entitled “The moderating role of Research and Development (R&D) support in the relationship between entrepreneurship and economic growth -A study on the GCC countries”. Despite the originality of topic, I’d recommend some major changes as follows:

- “Abstract”: I’d advice to reinforce and highlight originality, practical, and theoretical implications in it. Authors should better emphasize the research goals as well as the research design, placing more emphasis on the state of the art and on contributions of the paper.

- “Introduction”: I consider this section a bit convoluted and, for that, hard to follow. Firstly, starting from the beginning, I’d suggest Authors to better explain the focus of the research and to specify the scope of the paper. Secondly, please try to revamp the Introduction structure as follow: (i) define the contest of the analysis; (ii) clearly explain the gap in the literature that the paper wants to fill; (iii) point out the originality of the article (iv) describe the structure of the paper. Thirdly, I’d like to suggest Authors to better outline the scope of the research since from the Introduction section. Fourthly, certain unclear and long fragment sentences have affected the organization of research idea starting from the first paragraph. Please, revamp the whole section.

- “Literature Review”: In this section, it is recommended that Authors give a detailed discussion on each of the theoretical frameworks used and the relationship deduced from these frameworks to support this study. For this reason, it would be relevant to explore the themes of Research and Development (R&D) in the relationship between entrepreneurship and economic growth to understand the reason why Authors led to adopt this perspective of analysis.

Please, you can consider these international studies:

·      Scuotto, V., Magni, D., Palladino, R., & Nicotra, M. (2022). Triggering disruptive technology absorptive capacity by CIOs. Explorative research on a micro-foundation lens. Technological Forecasting and Social Change, 174, 121234.

- “Methodology” and “Data, results and discussion”: these sections appear well constructed. Well done!

- Discussions and Conclusion: Since I deem that the discussion is relevant to confute or support previous research, I’d reinforce this section properly. Yet, starting from the findings, I would suggest Authors to explain better the novelty of results and the main theoretical but also managerial implications of the paper. Alongside, please strengthen the discussion along with the rest of the article.

Quality of communication

The quality of communication is good. Nonetheless, a professional proof-reading would certainly increase the overall quality of the paper, thus meeting the international standards for peer-reviewed research.

I hope my advice will be useful for a further improvement of your paper.

Best Regards and Good Luck.

Author Response

Dear Reviewer

Thank you for taking the time to review our paper and for providing us with valuable feedback. We have carefully considered all the points you have raised and made the necessary changes to improve the quality of the paper.

We have reinforced and highlighted the originality, practicality, and theoretical implications in the abstract. We have also emphasized the research goals, the research design, the state of the art, and the contributions of the paper.

Regarding the introduction section, we have better explained the focus of the research and specified the scope of the paper. We have restructured the section to define the context of the analysis, clearly explain the gap in the literature, point out the originality of the article, and describe the structure of the paper.

In the literature review section, we have given a detailed discussion on each of the theoretical frameworks used and explored the themes of Research and Development (R&D) in the relationship between entrepreneurship and economic growth, as you suggested. We have also included the study you recommended (Scuotto et al., 2022).

We are pleased to hear that the methodology, data, results, and discussion sections appear well-constructed.

We have strengthened the discussion section, explained the novelty of the results, and provided better explanations of the main theoretical and managerial implications of the paper.

We have also taken note of your suggestion to have professional proofreading to increase the overall quality of the paper.

Thank you again for your valuable feedback, which has helped us improve the paper.

best regards,

Round 2

Reviewer 1 Report

Dear Authors,

thanks for the feedback and corrections of the article. I believe the quality of the article is now increased.

Best regards

Author Response

Dear Reviewer

Thank you for taking the time to review our article and for providing us with your valuable feedback. We appreciate your positive comments regarding the improvements made to the quality of the article.

Your input has been instrumental in enhancing the overall quality and clarity of our work. We hope that our article will contribute to the advancement of the field and address the key issues discussed in the paper.

Once again, we thank you for your time and attention, and we look forward to any further suggestions you may have.

best regards

Reviewer 3 Report

The paper does not merit publicaiton.

Author Response

Dear Reviewer,
Thank you for taking the time to review our paper. We appreciate your feedback and your concern regarding the merit of our work.
While we understand your position, we believe that our paper makes a valuable contribution to the field.
Once again, we appreciate your time and attention.
Best regards.

Reviewer 4 Report

Good luck

Author Response

Dear Reviewer

Thank you for taking the time to review our article. We appreciate your brief feedback and your wishes for our success.

We are constantly striving to improve the quality of our work, and your comments are valuable in this regard.
Once again, thank you for your time and attention.
best regards